

# Impact of climate change on wintertime European atmospheric blocking

Sara Bacer[1], Fatima Jomaa[1], Julien Beaumet[2], Hubert Gallée[2], Enzo Le Bouëdec[1], Martin Ménégoz[2], and Chantal Staquet[1]

[1]Univ. Grenoble Alpes, CNRS, Grenoble INP, LEGI, 38000 Grenoble, France
[2]Univ. Grenoble Alpes, CNRS, IRD, Grenoble INP, IGE, 38000 Grenoble, France

**Correspondence:** Sara Bacer (sara.bacer@univ-grenoble-alpes.fr)

**Abstract.** We study the impact of climate change on wintertime atmospheric blocking over Europe focusing on the frequency, duration, and extension of blocking events. These events are identified via the weather type decomposition (WTD) methodology applied on the output of climate models of the Coupled Model Intercomparison Project phase 6 (CMIP6). Historical simulations as well as two future scenarios, SSP2-4.5 and SSP5-8.5, are considered. The models are evaluated against the reanalysis and only a subset of climate models, which better represent the blocking weather regime in the recent-past climate, is considered for the analysis. We find that frequency and duration of blocking events remain relatively stationary over the 21st century. In order to quantify the extension of blocking events, we define a new methodology which relies on the WTD to identify blocking events. We show that the results are in agreement with previous studies that define blocking events with blocking indexes. We find that blocking extension will increase, especially in the worst-case scenario, due to a pressure increase driven by a thermodynamical warming during blocking events rather than atmospheric circulation changes.

## 1 Introduction

Atmospheric blocking is a persistent and quasi-stationary phenomenon which highly impacts the mid-latitude circulation. By obstructing the usual westerly winds, atmospheric blocking can promote cut-off cyclones (Munoz et al., 2020) and enhance cooling in winter and warming in summer. Its long duration (from days to weeks) affects surface weather and climate and fosters regional extreme events, such as heatwaves, droughts, and severe cold weather in winter (Barriopedro et al., 2010; Woollings et al., 2018, and references therein). Blocking events are generally associated with high-pressure systems. During anticyclonic periods, solar radiation and high temperatures in summer promote ozone formation, while thermal inversions with subsidence conditions in winter promote the accumulation of particulate matter (e.g. Largeron and Staquet, 2016; Hou and Wu, 2016).

Simulating blocking is a challenging task for atmospheric models as it requires an accurate description of the topography, a fine resolution both vertically and horizontally, appropriate physical parameterizations, and a correct description of internal dynamics (Davini et al., 2017). It has been shown that general circulation models (GCMs) are able to reproduce the blocking regime and its variability, although they tend to underestimate frequency and persistence of blocking events (Dunn-Sigouin





et al., 2013; Masato et al., 2013, 2014; Woollings et al., 2018; Davini and D'Andrea, 2020). Increasing model resolution can
improve the blocking occurrence, as the transient eddies and orography are better described (Berckmans et al., 2013; Schiemann
et al., 2017). Since atmospheric blocking is related to stratospheric variability (e.g. the stratospheric sudden warming, Davini
et al., 2014), a good representation of the stratosphere can also improve blocking simulations.

The identification of blocking events itself in numerical simulations is complicated by the fact that blocking is determined by
various dynamical mechanisms and presents different patterns. Several blocking indexes have been proposed in the literature,
based on meteorological fields, usually the geopotential height at 500 hPa (e.g. Tibaldi and Molteni, 1990), or anomalies of
meteorological fields (e.g. Dole and Gordon, 1983). Given the variety of blocking indexes, the comparison across studies is not
straightforward. Atmospheric blocking can also be identified as a weather regime (or weather type) via the so-called weather
type decomposition (WTD) methodology, consisting in the classification of atmospheric circulation into a certain number of
discrete regimes (Michelangeli et al., 1995). The WTD methodology, referred to as *the WTD* hereafter for brevity, relies on
a partitioning algorithm that groups data of a meteorological variable (usually geopotential height or sea level pressure) into
clusters so that the variance between clusters is maximized and the variance within the same cluster is minimized. In the
European-Atlantic sector, for example, four weather types have been recognised: positive North Atlantic Oscillation (NAO),
negative NAO, Atlantic ridge, and European blocking (e.g. Michelangeli et al., 1995; Cassou et al., 2004).

The impact of blocking events is related to their spatio-temporal characteristics, such as occurrence, duration, and extension.
Many studies investigated frequency and duration of blocking events in the past climate using reanalysis data (e.g. Wiedenmann
et al., 2002; Barriopedro et al., 2006; Mokhov et al., 2013; Cheung et al., 2013; Drouard and Woollings, 2018; Lupo et al.,
2019). Understanding the impact of climate change on atmospheric blocking is of fundamental importance to estimate future
climate and extreme events, thus, blocking has also been investigated in the future in response to global warming. For example,
it has been shown that the Arctic amplification, which has a strong influence on mid-latitude atmospheric circulation, modulates
the frequency and the intensity of blocking events (e.g. Hassanzadeh et al., 2014). Climate models suggest that blocking
frequency may decrease in the Northern Hemisphere in the future (e.g. Dunn-Sigouin and Son, 2013; Fabiano et al., 2020), and
blocking activity could shift eastwards (e.g. Masato et al., 2013, 2014; Woollings et al., 2018), while there is no clear tendency
for changes in blocking duration.

So far, studies have mainly focused on frequency and duration changes of future blocking events (e.g. Barriopedro et al.,
2006; Patterson et al., 2019; Lupo et al., 2019). Future changes in blocking extension have received less attention (Nabizadeh
et al., 2019). These works determined blocking events via blocking indexes and considered one or more GCMs participating in
the Coupled Model Intercomparison Project phase 5 (CMIP5). To our knowledge, only Fabiano et al. (2020) employed CMPI6
models in order to project future weather types and analyse their changes in frequency and duration.

In this study, we investigate the impact of climate change on European atmospheric blocking in terms of frequency, duration,
and especially extension. Several GCMs of the latest model intercomparison CMIP6 are considered for this purpose under two
different future scenarios (SSP2-4.5 and SSP5-8.5). In order to identify blocking events, the WTD is applied. We focus on
wintertime blocking as it is more frequent, longer, and stronger than blocking in summer in the European-Atlantic sector
(Barriopedro et al., 2006; Cheung et al., 2013; Lupo et al., 2019). Moreover, winter blocking events are often associated with





severe particulate matter pollution episodes. We introduce a new method, referred to as *the center method*, to quantify the extension of blocking events that are identified via the WTD. We compare the results obtained with this method with the results obtained for the blocking events identified via the index of Dole and Gordon (1983). Besides using GCMs of the latest CMIP phase, investigating frequency, duration, and extension of blocking events that are determined via the WTD instead of blocking indexes makes this work an original study.

## 2   Data

Daily means of geopotential height at 500 hPa (Z500) are used for the WTD. More precisely, the WTD is applied on winter anomalies of Z500, where the winter season is defined from 1 November to 31 March (NDJFM, like in Cassou (2008), for instance). The numerical domain of Z500 covers the European-Atlantic sector whose boundaries are $80°W$, $50°E$, $20°N$, and $80°N$.

In this study, GCMs of the CMIP6 (Eyring et al., 2016) are considered. It has been shown that the weather regimes are reproduced better in CMIP6 models than in CMIP5 models, especially over the European-Atlantic sector (Fabiano et al., 2020; Davini and D'Andrea, 2020). We use historical runs to analyse blocking conditions in recent-past climate and two future projections, SSP2-4.5 and SSP5-8.5 (Riahi et al., 2017), to investigate their changes in future climate. SSP2-4.5 assumes that social, economic, and technological trends broadly follow their historical patterns and is considered as a likely scenario given the current policies. In contrast, SSP5-8.5 projects strong increments of emissions without mitigation policies; it is the worst-case scenario and is considered unlikely (Hausfather and Peters, 2020). We also use the ERA5 reanalysis of the European Centre for Medium-Range Weather Forecasts with a resolution of 31 km (Hersbach et al., 2020) to evaluate the GCM ability in reproducing the blocking weather regime.

The Z500 outputs considered in this study are archived in the Mésocentre ESPRI. We selected the nine CMIP6-GCMs presented in Table 1 according to the following criteria: one GCM per each climate research centre, as different versions of the same model could present model-dependent similarities (Ullmann et al., 2014); GCMs having both SSP2-4.5 and SSP5-8.5 scenarios available; GCMs with the "r1i1p1f1" run available (where "r1": initial conditions, "i1": initialization method, "p1": physical scheme, and "f1": forcing configuration), as this is the most frequently accessible simulation. The analysed periods are 30-year long: 1980-2009 (HIST hereafter) and 2070-2099 (SSP2 or SSP5 hereafter, according to the scenario).

## 3   Methods

### 3.1   Detection of the blocking weather regime

In this study, blocking events are identified through the application of the WTD. This weather classification has largely been used in order to infer the recurrent atmospheric features at mid-latitudes (Michelangeli et al., 1995; Philipp et al., 2016). It can be divided into two steps: dimensional reduction of the data set and clustering. Similarly to other studies (e.g. Boé and Terray, 2008; Hertig and Jacobeit, 2014; Sáenz and Durán-Quesada, 2015), we apply the Principal Component Analysis (PCA)





| Climate center | GCM | Acronym | Lon x Lat |
|---|---|---|---|
| Beijing Climate Center (China) | BCC-CSM2-MR | BCC | 1.1°x1.1° |
| Canadian Centre for Climate Modelling and Analysis (Canada) | CanESM5 | CanESM | 2.8°x2.8° |
| Institute of Atmospheric Physics (China) | FGOALS-g3 | FGOALS | 2.0°x2.0° |
| National Oceanic and Atmospheric Administration (USA) | GFDL-CM4 | GFDL | 2.5°x2.0° |
| Institute of Numerical Mathematics (Russia) | INM-CM5-0 | INM | 2.0°x1.5° |
| Institut Pierre-Simon Laplace (France) | IPSL-CM6A-LR | IPSL | 2.5°x1.3° |
| Atmospheric and Ocean Research Institute (Japan) | MIROC6 | MIROC | 1.4°x1.4° |
| Max Planck Institute for Meteorology (Germany) | MPI-ESM1-2-HR | MPI | 0.9°x0.9° |
| Meteorological Research Institute (Japan) | MRI-ESM2-0 | MRI | 1.1°x1.1° |

**Table 1.** The CMIP6-GCMs used in this study. The columns contain, respectively, the name of the research center developer of the GCM, the name of the GCM, the acronym used in this study, and the resolution of the Z500 output. All data were provided by the Mésocentre ESPRI.

for the first step and the k-means algorithm for the second step. After the PCA, the eigenvectors necessary to explain 95% of the total variance (24 eigenvectors on average) are retained to define the reduced data set. k-means is applied on this data set by imposing that the number of clusters (k) is equal to four, i.e. the four well-known weather types of the European-Atlantic sector (positive and negative NAO, Atlantic ridge, and European blocking), as done in Cassou (2008), Ullmann et al. (2014), and Fabiano et al. (2020).

95       For ERA5 and each GCM of Table 1 and for each period (HIST, SSP2, and SSP5) the following procedure is followed. First, daily anomalies of Z500, noted $\Delta Z500$, are computed as difference between the 30-year daily means of Z500 and the annual cycle of the 30-year period used as a climatology reference; only the winter season (NDJFM) is retained. Second, the anomalies are weighted (multiplied) by the square root of the cosine of the latitude (Chung and Nigam, 1999) in order to account for the convergence of the meridians and so decrease the impact of high-latitude grid boxes that represent a small

area of the globe (like in Cassou, 2008; Ullmann et al., 2014; Cortesi et al., 2019). Since the GCMs have different resolutions (Table 1), the anomalies are linearly interpolated onto a common grid of resolution 1°x1°. Then, the PCA is applied to the resulting anomalies. Finally, k-means is performed, and each day of HIST, SSP2, and SSP5 is assigned to one of the four weather types. Only the weather regime corresponding to the European atmospheric blocking is analysed in this study.

        Climate change impact on blocking is quantified with respect to the historical reference period (like in Cattiaux et al., 2013;

Davini and D'Andrea, 2020). This means that the HIST annual cycle is used as a climatology to compute the anomalies ($\Delta Z500_{HIST}$) in all periods (HIST, SSP2, SSP5). However, in order to understand if the spatio-temporal characteristics of blocking will change in the future because of a modified atmospheric dynamics rather than warming climate, we also compute the future anomalies by subtracting the corresponding SSP annual cycle; these anomalies are noted $\Delta Z500_{SSP2}$ and $\Delta Z500_{SSP5}$. Therefore, the comparison between past and future $\Delta Z500_{HIST}$ will show the impact of the total climate change signal, governed

by greenhouse gases increase, global warming, and associated regional circulation changes. On the other hand, the comparison between past $\Delta Z500_{HIST}$ and future $\Delta Z500_{SSP}$ will show the blocking changes with respect to the climatology of that period;





this will allow to quantify the dynamical signal ignoring the thermodynamical signal related to the anthropogenic warming. Both $\Delta Z500_{HIST}$ and $\Delta Z500_{SSP}$ undergo the same analysis process; since the final aim of this work is to investigate the net climate change impact on blocking, the results obtained with $\Delta Z500_{SSP}$ that are similar to the $\Delta Z500_{HIST}$ results are included in the supplementary material.

## 3.2 Definition of blocking events

Consecutive days that belong to the blocking weather type can form a blocking event. The blocking events considered in this study satisfy the following conditions: they must be longer than five days (like in Barriopedro et al., 2006; Matsueda et al., 2009; Mokhov et al., 2013) and separated by at least two non-blocking days. A single non-blocking day ("hole") is assumed to represent a failure of k-means in that day, like in Matsueda et al. (2009). Therefore, two blocking events longer than two days separated by a hole form one blocking event; one blocking day and one blocking event longer than three days separated by a hole form one blocking event.

We clarify here the meaning of some specific terms. We call *centroids* the four centres of mass defined by the k-means algorithm in the reduced space, i.e. the space whose coordinates are the eigenvectors. Weather regime (or weather type) refers to the centroid transformed into the original latitude-longitude coordinate space. From now on, we call *blocking days* those days which belong to a blocking event. Finally, we refer to the temporal mean of $\Delta Z500$ over the blocking days of a blocking event as the *composite* of that event.

## 3.3 Computation of blocking area

We quantify the extension of a blocking event by its area. Two distinct methods are used to compute the blocking area: the so-called center method, introduced in this study, and the *DG method*, used by Nabizadeh et al. (2019).

**Center method.** We introduce this method to compute the area of the composites of the blocking events inferred from the WTD. The center method starts from the detection of the center of each blocking event. In this study, we define as *center* of the European atmospheric blocking the location of the maximum positive anomaly of the composite between 30°W and 50°E (similarly to Barriopedro et al., 2006), in order to discard blocking events with positive anomaly on the westernmost part of the sector. The extension of the blocking event is quantified by the area enclosed within the contour line equal to a certain threshold. In order to get non vanishing areas, we define a threshold of $\Delta Z500$ that must be lower than the minimum value among the centers over all periods and all GCMs. Technical details about the center method are reported in the Supplement.

**DG method.** This method follows the work of Nabizadeh et al. (2019), who determined the extension of the atmospheric blocking events identified with the index of Dole and Gordon (1983) (DG index hereafter). First, they compute the daily DG index for each grid box of the domain. Second, they identify as blocking events those grid boxes where the DG index is higher than 1.5 for at least five consecutive days. We will refer to these days as *DG-blocking days* and to the blocking events as *DG-blocking events*. Then, for each DG-blocking day, they compute the area enclosed by the contour line where the DG index is equal to one (i.e. the contour line equal to a certain threshold of $\Delta Z500$). Finally, daily areas are averaged along the event duration to get the area of the DG-blocking event (more details about the DG method are in the Supplement). In the present





paper, the DG-blocking days will be identified within the blocking events inferred from the WTD (and therefore not during the entire winter, as considered by Dole and Gordon (1983)).

The blocking areas presented in subsection 4.4 will be computed via both the center method and the DG method. As previously mentioned, a certain threshold is defined in both methods to delimit the blocking extension. For the purpose of this study, which requires a comparison between past and future results among several GCMs, we keep the thresholds constant for
the entire analysis.

## 4 Results and Discussion

### 4.1 Evaluation of the GCMs

Before analysing the impact of climate change on European atmospheric blocking events, the ability of the GCMs (Table 1) in reproducing the blocking weather regime is evaluated with respect to the reanalysis with a Taylor diagram (Figure 1). This
diagram compares the blocking composites of each GCM during HIST with the ERA5 composites. The deviation is quantified in terms of pattern correlation ($R$), standard deviation ($\sigma$), and root-mean-square difference (RMSD). All GCMs are able to represent blocking variability (i.e. $\sigma$) quite close to the variability obtained with the reanalysis ($\sigma_{ERA5} \cong 61$ m). More precisely, the variability of all GCMs is within the range $\sigma_{ERA5} \pm 6$ m, apart from INM and IPSL. Six models (MPI, BCC, MRI, GFDL, INM, and FGOALS) show a high correlation ($R \geq 0.79$) with ERA5, while three models (CanESM, MIROC,
and IPSL) present a lower correlation ($R < 0.6$) and a high RMSD. Hertig and Jacobeit (2014) also found that historical runs of CanESM cannot well reproduce the blocking pattern, getting a correlation with reanalysis lower than $0.4$. In this study, CanESM is the GCM with the coarsest resolution (Table 1), and it has been shown that a low resolution hinders a good description of the atmospheric variability patterns (Berckmans et al., 2013).

This analysis points out that MIROC, IPSL, and CanESM are less accurate in capturing the blocking pattern in recent-past
climate (as also observable in Figure 2), and we expect these models to be less reliable in future projections of blocking. Previous studies (e.g. Chhin and Yoden, 2018; Mokhov and Timazhev, 2019; Khan et al., 2020) suggest to use a subset of GCMs selected according to their ability in simulating the quantity of interest (atmospheric blocking in this study) in the past in order to reduce the uncertainties associated to the future projections of that quantity. Therefore, we exclude MIROC, IPSL, and CanESM from the next analysis and focus on the results obtained by the other six GCMs: MPI, BCC, MRI, GFDL, INM,
and FGOALS. In the same Taylor diagram (Figure 1), blocking projected for future climates (both SPP2 and SSP5) by these six GCMs is also shown. Overall, correlation coefficients, standard deviations, and RMSDs vary in a non-systematic way, so we do not find any regularity in the reproducibility of future blocking by the GCMs.

The spatial patterns of blocking during recent-past climate are shown in Figure 2. All GCMs are considered during HIST, and the dissimilarity of CanESM, MIROC, and IPSL with respect to ERA5 is evident. According to the reanalysis, the European
blocking is centred over the Scandinavian peninsula and extends over northern Europe. Blocking occurrence is about 27% (Table S1 in the Supplement) in accordance with previous studies that considered, for example, NCEP/NCAR reanalysis (27%, Cassou, 2008) and ERA-interim reanalysis (26%, Ullmann et al., 2014). MPI, BCC, and MRI reproduce an occurrence similar





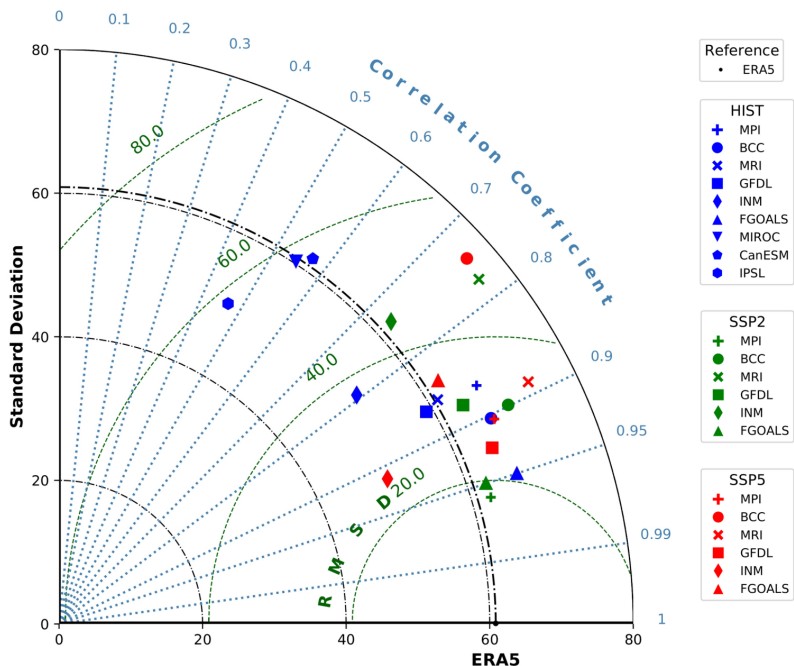

**Figure 1.** Taylor diagram for the mean composites over all blocking events for ERA5 and all GCMs for the winter HIST period (1980-2009). The diagram allows to quantify standard deviation (black), correlation coefficient (light blue), and root-mean-square difference (green) between the mean GCM composites and the mean ERA5 composite. SSP2 and SSP5 results are obtained with $\Delta Z500_{\text{HIST}}$.

to ERA5, while GFDL, INM, and FGOALS simulate less frequent blocking with an occurrence of about 23%. We observe that the first three models have the highest resolution (see Table 1), so we also find that the underestimation of atmospheric blocking

occurrence is reduced in high resolution GCMs. We compute the multi-model (MM) mean as the average of the composites over all blocking events of the six selected GCMs. The spatial pattern of the MM mean in HIST is very close to the ERA5 blocking, as also demonstrated by the statistics: $R \cong 0.98$, RMSD$\cong 13$ m, and $\sigma_{MM} \cong 56$ m.

In future climate (SSP2 in Figure 3 and SSP5 in Figure S2), the most evident change in blocking obtained with $\Delta Z500_{\text{HIST}}$ is the extension, which gets wider in SSP2 and especially in SSP5. Moreover, the centers of future blocking are characterised by

higher values of anomalies in comparison with ERA5; also in this case, the changes are emphasised in SSP5. On the contrary, analysing $\Delta Z500_{\text{SSP}}$ we find that the spatial patterns of the future blocking composites (Figure S3) are very similar to the results obtained for the HIST period (all these results will be confirmed in subsections 4.3 and 4.4). The strong differences in blocking extension and intensity between $\Delta Z500_{\text{HIST}}$ and $\Delta Z500_{\text{SSP}}$ results indicate that future blocking changes are mainly due to thermodynamical than dynamical changes. Thus, we find that atmospheric blocking presents a dynamical component

whose pattern is relatively stationary over the 21st century and a thermodynamical component that evolves in relation with the anthropogenic signal.

**Figure 2.** $\Delta Z500_{HIST}$ composites averaged over all blocking events for all GCMs and ERA5 during the winter HIST period (1980-2009); in the last row, the multi-model mean is computed over all blocking events of the six selected GCMs.

## 4.2 Frequency and duration of blocking events

Blocking events are identified for each GCM following the definition in subsection 3.2. The number of blocking days and blocking events per winter averaged over all winters of the 30-year periods and the duration of blocking events averaged over

these periods are graphically represented in Figure 4 to facilitate the comparison of the HIST results against reanalysis and the future (SSP2 and SSP5) results. We find that, during recent-past conditions, the MM mean number of blocking days per winter is about 30 and the MM mean number of events per winter is about 3 (Figure 4 and Table S2). Our results are slightly lower

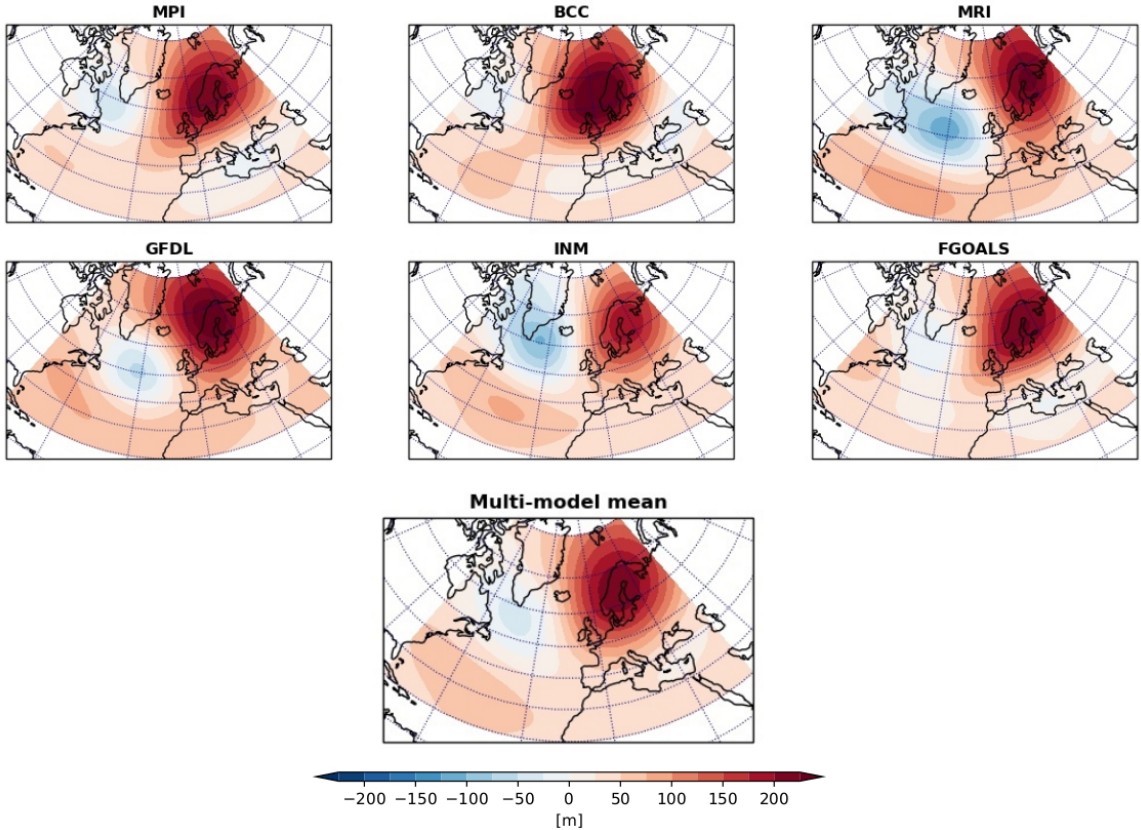

**Figure 3.** $\Delta Z500_{HIST}$ composites averaged over all blocking events for the six selected GCMs during the winter SSP2 period (2070-2099); in the last row, the multi-model mean is computed over all blocking events of the six selected GCMs.

than the findings of Mokhov et al. (2014), 35.8 days and 4.7 events, who detected blocking events in an Euro-Atlantic sector using a Z500-based blocking index applied on one GCM (IPSL). The MM mean of blocking duration is $9.9 \pm 0.9$ days and is close to the mean duration of blocking events of $10.2 \pm 5.3$ days obtained with the reanalysis. These results are in agreement with mean blocking durations found in the literature, e.g. 10.5 days for winter blocking in the European-Atlantic sector by Lupo et al. (2019), using reanalysis (NCEP/NCAR) of Z500, and 7.6 days by Mokhov et al. (2014). In summary, the mean characteristics of blocking events are well reproduced by the GCMs: the MM means of number of blocking days and duration during HIST are close to the results obtained with the reanalysis, although most of the models tend to underestimate these quantities.

When analysing the impact of climate change, no significant impact is found on blocking frequency and duration. With respect to the HIST results, MPI, BCC, and MRI simulate less frequent blocking events in both future scenarios, while the other GCMs present a higher blocking frequency (Figure 4). Additionally, results for SSP2-4.5 and SSP5-8.5 are not in agreement among the various models (sometimes estimates are higher in SSP2 and sometimes in SSP5). Nevertheless, we observe that



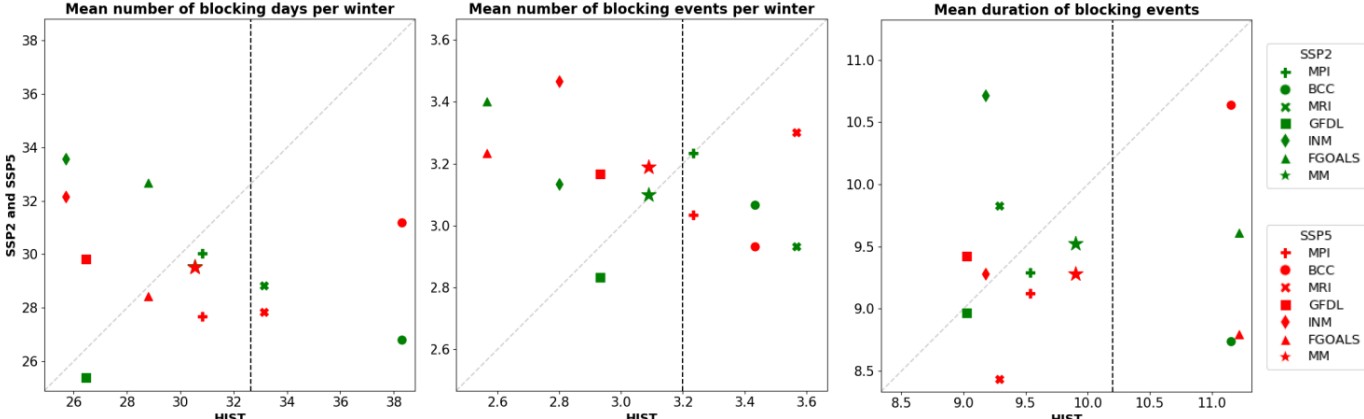

**Figure 4.** Number of blocking days (*left*) and blocking events (*center*) averaged over all winters of the 30-year periods, and mean duration (in days) of blocking events occurred in 30 winters (*right*) for recent-past climate and future scenarios (SSP2-4.5 and SSP5-8.5) considering $\Delta Z500_{HIST}$. The black dashed line is the ERA5 mean. (The values are taken from Table S2.)

the MM means of the mean number of blocking days per winter and the mean duration of blocking events will decrease by about one day and half a day, respectively. Interestingly, the results obtained with $\Delta Z500_{HIST}$ are very similar to the results obtained with $\Delta Z500_{SSP}$ (Figure S4). This suggests that the tendency of blocking frequency to decrease in the future is due to changes of the atmospheric dynamics under the SSP2-4.5 and SSP5-8.5 scenarios. However, the uncertainty of the results is very large (Table S2), so the differences between the periods are not statistically significant. Actually, a clear long-term

change in blocking frequency in the past has not emerged so far (Barnes et al., 2014; Woollings et al., 2018), and there is no general consensus on the tendency of blocking frequency in future climate (Woollings et al., 2018). For example, and Matsueda and Endo (2017) found a significant decrease in blocking frequency in the European-Atlantic sector involving all durations of blocking events simulated with six CMIP5-GCMs, Mokhov et al. (2014) found a general increase in the blocking frequency, while Masato et al. (2014) found that European blocking frequency remains unchanged using four CMIP5-GCMs.

The analysis of the occurrence of blocking events as a function of duration also indicates that the GCM projections agree well with the reanalysis (Figure 5 and Figure S5). Occurrence of blocking events decreases exponentially with duration, consistent with the findings of Wiedenmann et al. (2002); Barriopedro et al. (2006); Matsueda et al. (2009); Dunn-Sigouin and Son (2013); Mokhov et al. (2014). The distributions of all periods show long tails up to 30 days, but some isolated events can be even longer. In future climate (both SSP2 and SSP5), we find that the occurrence of short (5-8 days) blocking events increases, while the

occurrence of long (more than 10 days) events tends to decrease, as indicated by the mean lifetimes ($\tau$) of the exponential fits, which are lower for SSP2 and SSP5 ($\tau \approx 9$ days) than for HIST ($\tau \approx 10$ days). It must be noted that these results, obtained with $\Delta Z500_{HIST}$, are very close to the $\Delta Z500_{SSP}$ results (Figure S6).

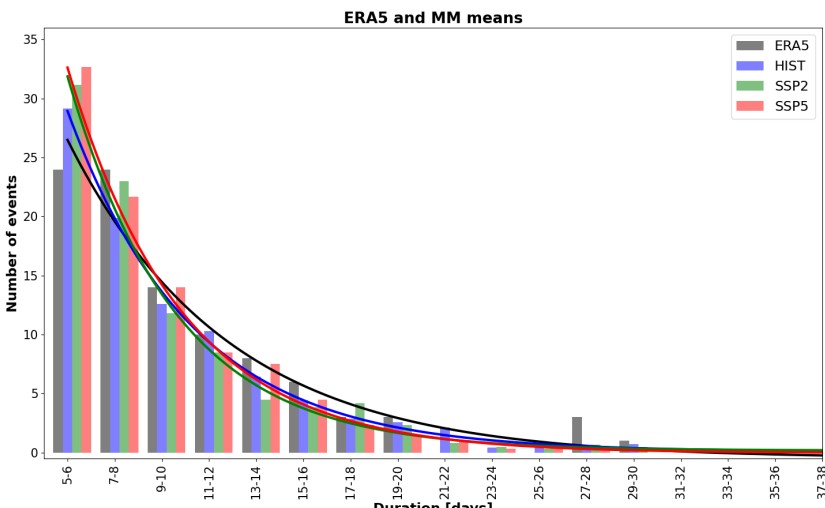

**Figure 5.** Occurrence of blocking events as a function of duration for ERA5 and MM means during HIST, SSP2, and SSP5 considering $\Delta Z500_{\text{HIST}}$. Exponential fits are drawn for ERA5 and MM means.

## 4.3 Centers of blocking events

We now analyse the blocking centers (as defined in subsection 3.3) of the composites of blocking events. We study the impact

of climate change on the blocking centers in terms of their location and intensity, i.e. the value of $\Delta Z500$ at that location. The geographical distribution of the center locations averaged over all blocking events of a given 30-year period is shown in Figure 6. The ERA5-center is located over Sweden. The GCM-centers during HIST are over and close to the Scandinavian peninsula. In the future, we observe a general eastward shift of the center locations using $\Delta Z500_{\text{HIST}}$. In particular, four out of six models during SSP2 and all models during SSP5 show blocking centers that are eastward with respect to the centers in

HIST. The SSP2- and SSP5-MM means of the center locations are located about 6° and 9° eastward to the HIST-MM mean, respectively. An eastward shift of European blocking would lead to an increase of blocking over Western Russia (Dunn-Sigouin and Son, 2013). More uncertain is the meridional shift of the centers in the future (three GCMs, GFDL, MPI, and INM, show a northward shift). Similar considerations are also valid for $\Delta Z500_{\text{SSP}}$ (Figure S7), although the eastward shift tendency is even less evident (between 4° and 5° in both future scenarios). An eastward and northeastward shift of European blocking was also

found by Masato et al. (2013, 2014) and Sillmann and Croci-Maspoli (2009), respectively. However, it must be stressed that there is a large variability associated to the blocking center locations on both meridional and zonal directions (as attested by the error bars in Figure 6), and the shift of the centers is not significant.

The MM mean of the blocking center intensities during HIST is $248 \pm 18$ m, very close to the ERA5-intensity, $251 \pm 48$ m (Table S3). The minimum intensity is simulated by INM, $219 \pm 50$ m, the maximum one by MPI, $274 \pm 61$ m. Under the

SSP2-4.5 and SSP5-8.5 scenarios, the MM means of the intensities increase with respect to the recent-past conditions in both future scenarios, especially in the worst-case scenario (Figure 7, left). In particular, they increase up to 306 m in SSP2 and



344 m in SSP5. Although the variability increases as well (see the standard deviations in Table S3), the intensity increments (of SSP2 versus HIST and SSP5 versus HIST) are significant. Mokhov et al. (2014) also found a tendency for an increasing blocking intensity in the European-Atlantic sector in the 21st century, in winter, by analysing similar scenarios (RCP2.6 and

RCP8.5). Moreover, we observe that the minimum intensities of all GCMs are higher in SSP2 than in HIST, and even higher in SSP5; this is valid also for the maximum intensities of almost all models (Table S3). The significant increase of the center intensities (i.e. of the geopotential height) found by all GCMs in the future considering $\Delta Z500_{HIST}$ is mainly explained by the general warming related to the anthropogenic greenhouse gases emissions occurring under the considered scenarios. On the contrary, the center intensities of the future blocking events identified using $\Delta Z500_{SSP}$ are very similar, on average, to the

intensities of the blocking events in recent-past conditions (Figure 7, right and Table S3), implying that the future changes of blocking intensity will be not affected by atmospheric dynamical changes.

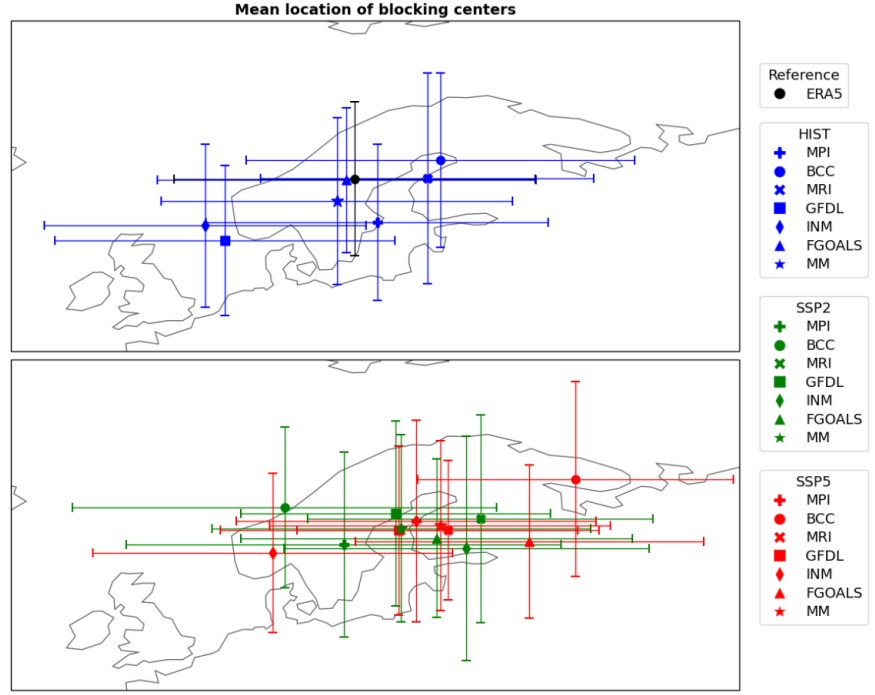

**Figure 6.** Locations of the blocking composite centers averaged over all blocking events for ERA5 and the GCMs during HIST (*top*), SSP2, and SSP5 (*bottom*) considering $\Delta Z500_{HIST}$. The error bars indicate the standard deviations of latitudinal and longitudinal coordinates of the blocking centers.





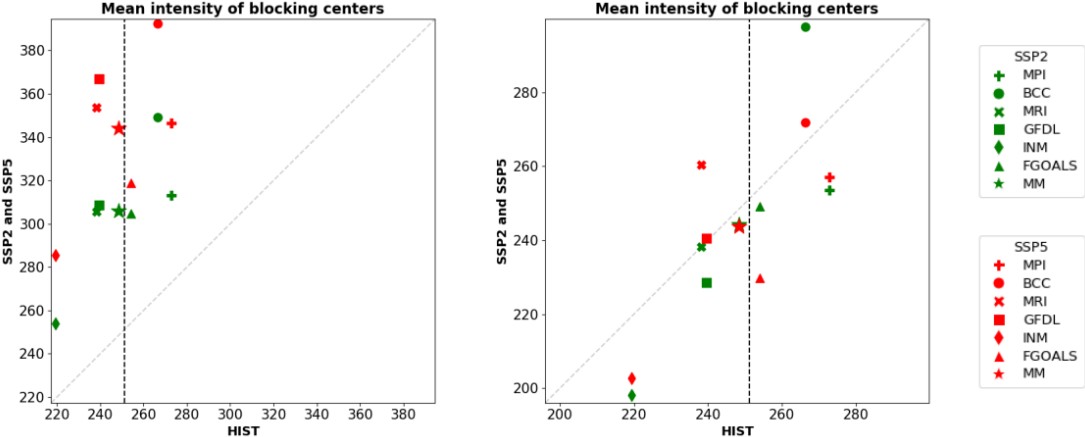

**Figure 7.** Intensities (in m) of the blocking composite centers averaged over all blocking events during HIST, SSP2, and SSP5 considering $\Delta Z500_{\text{HIST}}$ (*left*) and $\Delta Z500_{\text{SSP}}$ (*right*). The black dashed line is the ERA5 mean. (The values are taken from Table S3.)

## 4.4 Extension of blocking events

### 4.4.1 Center method

Blocking area is computed for each composite of blocking event by using the center method described in subsection 3.3.

This method takes into account events whose center is between $30°$W and $50°$E to focus on European blocking, nevertheless, some events can extend westwards in the European-Atlantic sector. This is due to the fact that the events have been identified via a partitioning algorithm (k-means) and not via blocking indexes designed for geopotential fields that are typical during atmospheric blocking. We could verify that, on average, only four events per GCM (i.e. $\sim 4\%$) are of this type during HIST. This effect is more frequent considering $\Delta Z500_{\text{HIST}}$ in the future, especially in SSP5, as the threshold more often allows for

the detection of "stretched" shapes. We preferred not to disregard them in order not to introduce subjectivity into the analysis, and the results are considered as an overestimation of the blocking extension, especially in the future.

The MM mean extension in HIST is $7.0 \cdot 10^6$ km$^2$, very close (1.4% larger) to the value obtained for ERA5 (Table S4). This extension is nearly twice (1.7) and three times (2.7) larger for SSP2 and SSP5. As expected from Figures 2, 3, and S2, we find a clear tendency of blocking extension to increase in the future, especially in the worst-case future scenario (in agreement

with Nabizadeh et al., 2019). The fact that positive anomalies get larger in the future is mainly due to the global warming projected with the SSP2-4.5 and SSP5-8.5 scenarios, as already noted in subsection 4.1. Different results are obtained for the future blocking extensions computed with $\Delta Z500_{\text{SSP}}$. In this case, the MM mean extensions in SSP2 and in SSP5 are similar (Table S4). Moreover, they are comparable to the blocking area in recent-past conditions (as anticipated in Figure S3).

We further analyse the blocking extension results in relation to the center intensity. We find a linear relation between exten-

sion of blocking events and intensity of blocking centers considering both $\Delta Z500_{\text{HIST}}$ and $\Delta Z500_{\text{SSP}}$ (Figure 8). The correlation is significant and higher than 0.8 in the HIST and SSP2 cases. The linear relation is in agreement with Barriopedro et al. (2010).





Again, our results are in line with previous studies that followed a different approach for the blocking detection, based on the use of blocking indexes instead of the WTD.

We observe that the blocking extension estimated for the GFDL model during SSP5 using $\Delta Z500_{HIST}$ is much higher than
the other GCMs (Figure 8 and Table S4). The reason is that the blocking events in GFDL are characterised, on average, not only by a large positive anomaly over North Europe, but also by an evident positive anomaly over the Atlantic Ocean close to the US coast (Figure S2). As a consequence, the shape of what is considered blocking extension in the center method is often elongated until North America (not shown). Since this type of shape cannot be attributed to the European blocking, the previous analysis has been performed also neglecting the GFDL model. In this case, the linear regression between blocking
extension and center intensity during SSP5 is higher than $0.8$ as well (Figure 8, left), thus, the GFDL results for SSP5-8.5 have not been considered in the next analysis with $\Delta Z500_{HIST}$.

Figure 9 shows that blocking extension is characterised by a normal distribution (e.g. Whiteman, 1982; Barriopedro et al., 2006). The similarity between the distribution obtained with reanalysis and historical GCM data is noteworthy (Figure 9 and Figure S8). The blocking extensions obtained with future $\Delta Z500_{HIST}$ can be only roughly approximated by a gaussian low,
especially in SSP5 (Figure 9, left). The results obtained in this case show that climate change will impact the distribution of future blocking area: more blocking events with larger extension will occur, as proved by the shift of the distribution towards higher values and its increasing width (although we remind that blocking area in the future is likely overestimated in this study). Different results are obtained for future blocking extensions computed with $\Delta Z500_{SSP}$ (Figure 9, right). In this case, the MM means of blocking area during SSP2 and SSP5 are very similar and close to the past blocking extension, as already inferred
with Figure S3. It must be specified that, since some values of center intensities are smaller than the threshold chosen for the $\Delta Z500_{HIST}$ case (100 m), the threshold used for the center method in the $\Delta Z500_{SSP}$ case is different (75 m for all periods, HIST, SSP2, and SSP5, see the Supplement for more details).

### 4.4.2   Comparison with the DG method

In order to check the reliability of the center method in estimating the area of blocking events, we compute that area by
another approach relying on the DG index to identify blocking events, as done by Nabizadeh et al. (2019). As indicated in subsection 3.3, the latter events will be denoted DG-blocking events; for clarity, in this section, the blocking events identified by the WTD will be denoted WTD-blocking events.

As explained in subsection 3.3 and in the Supplement, we apply the DG method to compute the area of those DG-blocking days that belong to the WTD-blocking events. Despite the number of DG-blocking days may not match with the duration of
the WTD-blocking events, we find that it agrees well with the duration of the WTD-blocking events. Such agreement improves from HIST to SSP2 and to SSP5 considering $\Delta Z500_{HIST}$. This is due to the fact that the positive anomalies get wider in SSP2 and even more in SSP5, the DG index is higher than 1.5 more often, and thus the DG-blocking events embrace or well overlap the WTD-blocking events of the SSP2 and SSP5 periods.

The blocking areas resulted from the center method and the DG method are compared in Figure 10. These areas are linearly
correlated with statistical significance in all periods, the slopes of the linear regression being larger than 0.79 with $\Delta Z500_{HIST}$



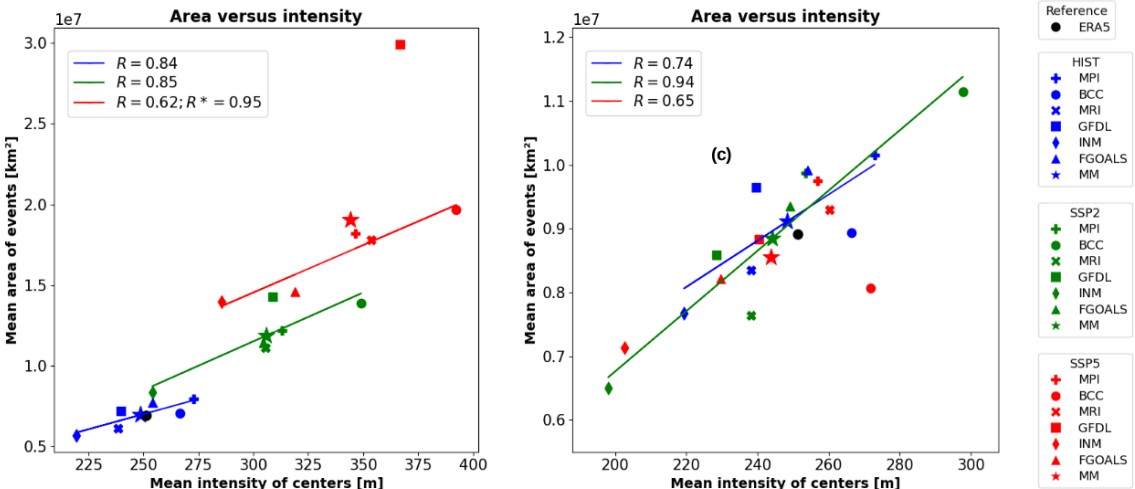

**Figure 8.** Mean area versus mean center intensity of blocking events computed for ERA5 and the GCMs during HIST, SSP2, and SSP5 considering $\Delta Z500_{HIST}$ *(left)* and $\Delta Z500_{SSP}$ *(right)*. $R$ is the correlation; $R^*$ is the correlation excluding GFDL; regression lines (found by the least-squares fit excluding GFDL in SSP5 in *(left)*) are drawn when the correlation is statistically significant (at the 90% confidence level).

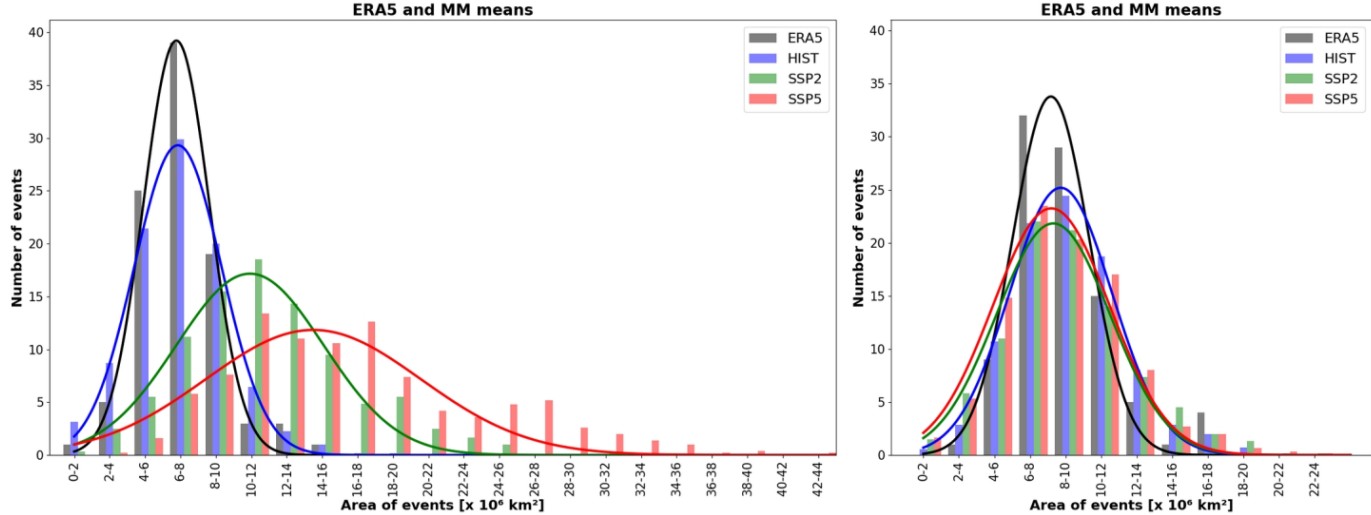

**Figure 9.** Occurrence of blocking events as a function of area for ERA5 and the GCMs during HIST, SSP2, and SSP5 considering $\Delta Z500_{HIST}$ *(left)* and $\Delta Z500_{SSP}$ *(right)*. Gaussian fits are drawn for ERA5 and MM means. GFDL is excluded from the MM means of SSP5 in $\Delta Z500_{HIST}$.

and 0.68 with $\Delta Z500_{SSP}$. We can conclude that the extension of the blocking events identified via the WTD is in agreement with the extension of the blocking events identified via the DG index.

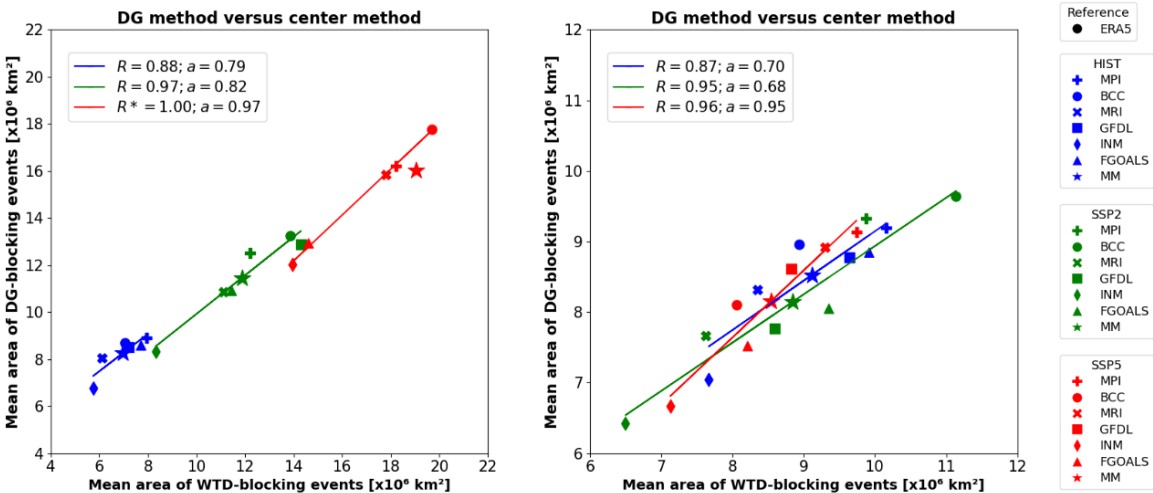

**Figure 10.** Mean area of blocking events computed with the DG method versus mean area of blocking events computed with the center method considering $\Delta Z500_{\text{HIST}}$ *(left)* and $\Delta Z500_{\text{SSP}}$ *(right)*. $R$ is the correlation; $R^*$ is the correlation excluding GFDL; $a$ is the slope of the linear regression; regression lines (found by the least-squares fit excluding GFDL in SSP5 in *(left)*) are drawn when the correlation is statistically significant (at the 95% confidence level).

## 5 Conclusions

We identify wintertime European blocking events by applying the weather type decomposition methodology on the European-315 Atlantic sector. Our aim is to quantify the impact of climate change on the frequency, duration, and extension of blocking events. For this purpose, we consider 30 years of historical runs and two future scenarios (SSP2-4.5 and SSP5-8.5) of nine CMIP6-GCMs. We show that the GCMs considered in this study capture well the spatio-temporal characteristics of atmospheric blocking, nevertheless, only those representing blocking patterns and variability closer to the reanalysis are used to investigate future blocking changes.

Considering two types of geopotential anomalies, which use a different climatology as a reference ($\Delta Z500_{\text{HIST}}$ and $\Delta Z500_{\text{SSP}}$), we can attribute the future changes of blocking to the total signal or to the dynamical signal of climate change. We find that the impact of climate change on blocking frequency and duration is not statistically significant, consistent with the literature results that there is no general consensus on the tendency of blocking event frequency in the future (Woollings et al., 2018). The fact that the results obtained for blocking event frequency and duration with $\Delta Z500_{\text{HIST}}$ are similar to the ones obtained 325 with $\Delta Z500_{\text{SSP}}$ suggests that changes in temporal characteristics of blocking are not only influenced by the global warming of the 21st century but also by changes in regional circulation.

We define a new methodology, the center method, to quantify the extension of blocking events. We find that blocking area and center intensity are linearly correlated. We apply another method, the DG method, and obtain similar results. This implies





that the area of a blocking event can be computed indifferently either by the WTD for the center method or by the DG index
for the DG method.

Climate change will significantly increase the extension of blocking events in the future especially in the worst-case scenario.
Blocking patterns and extension obtained with $\Delta Z500_{SSP}$ in the future are similar to the results obtained for the recent-past
climate. This means that the spatial characteristics of blocking events will not change at the end of the century with respect to
the climatology of the considered 30-year period and that the blocking extension increase due to climate change is mainly due
to higher geopotential height caused by warmer climate. Similar considerations are also valid for the mean intensities of the
blocking centers: they will increase during SSP2 and even more during SSP5 with respect to the recent-past conditions (using
$\Delta Z500_{HIST}$) because of the thermodynamical signal of climate change.

To the best of our knowledge, this is the first study investigating frequency, duration, and extension of blocking events
that are identified via the WTD. Moreover, there are still few studies addressing this topic using GCMs of the CMIP6. Our
results are in agreement with previous findings where blocking events are defined with blocking indexes. This confirms that
the application of the WTD is also a good strategy to analyse blocking event characteristics. This study could be improved
by analysing more GCMs, although other studies that considered many GCMs initially used only the best few GCMs for the
analysis later; for example, Lee and Ahn (2017) selected five GCMs among twenty-two CMIP5-GCMs to study atmospheric
blocking over the Pacific Ocean. Before comparing blocking event areas with other studies, it must be reminded that the results
depend on the defined threshold. Finally, it must pointed out that the four weather types imposed in the k-means algorithm
allow to recover the ones usually obtained with the reanalysis. However, a different number of weather types may need to be
computed in some models where the variability is different from the reanalysis (e.g. five regimes are considered in the CNRM
model by Ménégoz et al. (2018)).

*Author contributions.* All authors contributed to designing the study. SB and FJ analysed the data, together with ELB, and produced the
figures. All the authors discussed the results. SB wrote the paper; all the authors provided assistance in finalizing the article.

*Competing interests.* The authors declare that they have no conflict of interest.

*Acknowledgements.* The authors thank Dr. E. Nabizadeh and Prof. P. Hassanzadeh for the useful discussion. The authors acknowledge
the World Climate Research Programme (WCRP), which coordinated and promoted CMIP6, the climate modeling groups for producing and
making available their model output, and the Mésocentre ESPRI (Ensemble de services pour la recherche à l'IPSL, https://mesocentre.ipsl.fr/)
for archiving the data and providing the access to the CMIP6 data. SB thanks the LEGI for the postdoctoral support; ELB and FJ thank the
IDEX program MobilAir for the PhD and MSc support, respectively.





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
