# Peer review of "Impact of climate change on wintertime European atmospheric blocking"

_Weather and Climate Dynamics, 2021_

## Author Comment (AC1)

**Authors' replies to Referee #1**

We thank Referee #1 for their constructive comments. Below, we provide our replies; the line numbers and sections refer to the old version of the manuscript.

**Major Comments**

**1. WTD**

**1a. Discussion on advantages and limitations of WTD.** *I find no discussion in the paper about advantages and limitations of WTD (except description of how it is done and that it agrees with blocking indices). One of the limitations of WTD is that WTD is less flexible and it does not come with a native measure of blocking center and size.*

We agree with the Referee that such a discussion is currently not well developed. Following also the suggestions of Referee #2, we discussed pros and cons of the WTD but also of the blocking indexes in the Introduction. Please, find below the new text (in blue) added in paragraph 28-38.

"The identification of blocking events in numerical simulations is complicated by the fact that blocking is determined by various dynamical mechanisms and presents different patterns. Several blocking indexes have been proposed in the literature, based on meteorological fields, usually the geopotential height at 500 hPa (e.g. Tibaldi and Molteni, 1990), or anomalies of meteorological fields (e.g. Dole and Gordon, 1983). Blocking indexes focus on different characteristics of blocking, so the choice of the index depends on the purpose of the study. Additionally, index definitions depend on various (user-dependent) parameters, like latitude band limits, latitude references, and anomaly thresholds (a review of the blocking indexes can be found in Barriopedro et al. (2010), while a recent discussion about their differences is in Pinheiro et al. (2019)). Given the variety of blocking indexes, the comparison across studies is not straightforward.

Atmospheric blocking can also be identified via the so-called weather type decomposition (WTD) methodology, which classifies the atmospheric circulation into discrete weather regimes (Michelangeli et al., 1995). The WTD methodology, referred to as *the WTD* hereafter for brevity, relies on a partitioning algorithm that groups data of a meteorological variable (usually geopotential height or sea level pressure) into clusters so that the variance between clusters is maximized and the variance within a given cluster is minimized. In this way, the clusters (weather regimes or weather types) are the result of a mathematical algorithm. The results of the WTD depend on certain user choices, such as the sector size, the clustering algorithm and the initialization of this algorithm. Despite the fact that the clusters may not be well separated, WTD has proved to be very useful in the literature. In fact, WTD allows to explain most of the atmospheric variability and has largely been used to define weather regimes especially in the Northern Hemisphere (e.g. Michelangeli et al., 1995; Cassou et al., 2004; Barriopedro et al., 2006; Ullmann et al., 2014; Fabiano et al., 2020). In the European-Atlantic sector, for example, four winter weather types have been recognized: positive North Atlantic Oscillation (NAO), negative NAO, Atlantic ridge, and European blocking. The WTD has also been used to analyze weather types in relation to other quantities like temperature (e.g. Cassou et al., 2005), precipitations (e.g. Ullmann et al., 2014), winds (e.g. Jiménez et al., 2009), and

pollutants (e.g. Russo et al., 2014). In this study, the WTD is used to identify blocking events in the European-Atlantic sector."

**1b. Fitness of WTD.** *I wonder how well WTD can summarize the Z500 variability. In the process of k-mean, is variance between cluster much larger than variance within the same cluster? Or do the 4 clusters explain a very high percentage of variance? Or is the clustering very clear cut?*

There is a consistent number of studies showing that the WTD applied on Z500 can be used to divide the atmosphere at the synoptic scale into weather regimes, see the references cited in the Introduction and the new text written in point 1a above. Most of the studies considering the European-Atlantic sector in winter show that the clustering is optimized with k=4, so the WTD used to define the four weather regimes in this sector can be as a "standard procedure". The evaluation of the WTD methodology is beyond the purpose of this work, rather we evaluated the WTD results by comparing the four weather types with previous literature (section 4.1); since they agreed well with previous studies, we could develop our analysis on the basis of these WTD results. In the new version of the manuscript, we added a new Figure (please, see Fig. 1 in the replies to Referee #2) to show the agreement between the blocking episodes identified via the WTD and the DG-index.

**1c. Interpretation of WTD.** *If the WTD clustering is not that clear cut, it is hard for me to interpret the results. When "cluster centroid" is found different in some models, what does it mean? Is it because reanalysis-blocking-like pattern occurs less frequent? Or is it because some boundary cases (under reanalysis clustering) occur more frequent and that the cluster boundary needs to be put elsewhere?*
*Also the "blocking frequency". Do changes come from the 5-day requirement, or the overall frequency of weather type? If it has to do with the overall frequency of weather type, the question again is whether it comes from changes in frequency of centroid-like patterns or boundary-case-like patterns? Same question for the "blocking center". Does it come from the weather type shifting in location? If so, is it because of centroid-like patterns or boundary-case-like patterns?*

Differences of weather types among GCMs occur because global simulations depend on factors like internal climate variability, resolution, orography, and parameterizations that are GCM specific. Thus, there is the possibility that a GCM captures some recurrent patterns less well than other GCMs, affecting the WTD results (i.e. the weather types resulting from the clustering). The evaluation of the models in subsection 4.1 has exactly the aim of selecting those GCMs that well reproduce the weather regime of blocking, following the approach of other studies (cited in 4.1 subsection).

The frequency of the blocking days resulting from the clustering differs a bit (about 20%) from the frequency of the blocking days belonging to the WTD-blocking events. This is natural given the application of the criteria defined in subsection 3.2, e.g. the minimal duration of 5 days to define a blocking event. We would like to point out that such criteria are the ones used in the literature when blocking days are identified via blocking indexes, thus, we expect that also in that case the frequency of the blocking days is slightly different from the frequency of the blocking days belonging to the blocking events (although, we are not aware of any

quantification of this "frequency change" in the literature).

**1d. Insignificant results.** *If the WTD clustering is not that clear cut, I wonder if this will cause extra variability that stops you from drawing significant results. Most results using DeltaZ500_SSP are not statistically significant.*

We do not know if this "extra variability" is the cause of getting not significant results. We cannot exclude this, but we would like to point out that one could obtain not statistically significant results also with a clear cut of the clusters. Finding not statistically significant changes is a result itself, and this was the case also for other works cited in the manuscript (in fact, the review of Woollings2018 states that no clear long-term changes have been found in blocking frequency).

**2. DeltaZ500_HIST may be irrelevant.**

*Many results are based on DeltaZ500_HIST, in which the overall higher geopotential height in warmer climate is not removed. I cannot see how this overall higher geopotential height would link to weather impact or air pollution, which is likely why authors are interested in blocking. Measures of blocking based on DeltaZ500_HIST go too far, and become irrelevant to weather impacts.*

We agree with the Referee that if we analysed climate change impact on air pollution during blocking we should consider the results obtained with DeltaZ500_SSP. Following also the suggestions of Referee #2, we decided to present the DeltaZ500_SSP results in the revised version of the manuscript and to move most of the DeltaZ500_HIST results to the Supplement (or to remove them).

**Minor comments**

*3. Line 103: How do authors determine which (out of 4) weather regime is the European blocking weather regime?*

The four weather regimes obtained via the WTD in this study are close and comparable to the (usual) four weather regimes of the European-Atlantic sector described in previous literature. For instance, Figure 1 (below) shows the four weather types obtained with the ERA5 reanalysis in this study. We can observe that the association between the weather types (WTs) meant as the centroids resulting from k-means and the four atmospheric weather regimes is clear: WT1 is blocking, WT2 is Atlantic ridge, WT3 is NAO+, and WT4 is NAO-. Since the WTD was applied to each model, such association was defined each time.

[Figure]

Figure 1: The four weather regimes obtained with the WTD (and k = 4) over the European-Atlantic sector using ERA5 reanalysis (1980-2009).

*4. I prefer to say "size" or "area" of blocking, in place of "extension". The latter is not clear to me and let me think of temporal extension (duration), or some kind of extension of concept.*

We replaced "extension" with "size". The term "area" is used to quantify the blocking size, as written at line 129.

*5a. Line 131: I suggest change "center method" to "composite method". Because "composite" is really the step that differs from the "DG method".*

Ok, we replaced "center method" with "composite method".

*5b. But actually, the authors made a few modifications to the DG method that makes it very similar to the center method. For example, authors require DG blocking day to be a subset of WTD blocking day. Perhaps authors need to say what are the major remaining differences (if any) in the two method.*

We did not modify the procedure followed by Nabizadeh et al. 2019, a part from the fact that it is applied on WTD-blocking events (as the Referee pointed out), and we would not say that the two methods are similar. Although the algorithm to compute the blocking area is the same, it is applied on composites in the composite method and on daily anomalies in the DG-method. Moreover, the $\Delta Z500$ values of the contour lines (i.e. the thresholds) are defined in different ways: they depend on the center intensities in the composite method, while they depend on $\widetilde{\sigma}_{max}$ in the DG method. We made a few modifications in the Supplement (section "Computation of blocking area") to improve the description of the two methods and we highlighted the differences between the two methods at line 147:

"Therefore, these methods compute the area of blocking events that are identified via two

different approaches: the WTD and the DG index. Although the algorithm to compute the blocking area is the same, it is applied on blocking composites in the composite method and on daily $\Delta Z500$ in the DG method. Another difference between these two methods is the definition of the $\Delta Z500$ values of the contour lines (i.e. the thresholds)."

*6. Line 7: I think your methodology to quantify size of blocking does not "rely on the WTD". I don't think it is WTD-native or WTD-specific. I don't think this is sufficiently different from other studies (like Nabizadeh et al. 2019) that I would claim "new".*

We removed the adjective "new". We preferred to leave "rely on the WTD" as the center method is applied on WTD-blocking events, moreover, since we changed "center method" to "composite method" (as asked by the Referee), we think that this expression ("rely on") could remain in the Abstract.

*7. Line 9: "Geopotential height increase" might be more accurate than "pressure increase".*
Done.

*8. Explanations of methods are disordered. For example, the 3 paragraphs in section 3.1 go by talking first about WTD, then an overview of all steps, and lastly the calculation of Z500 anomaly (which is done before WTD).*

We changed the first two paragraphs in subsection 3.1 in order to explain the methodology step by step. We moved the third paragraph in a new subsection with title "Z500 anomalies" to explain the meaning of $\Delta Z500$.

*9. Line 105: I would suggest to add a bracket "(including the mean)" after "annual cycle". Because "annual cycle" can sometimes only refer to the seasonal variation from the mean.*

We made this addition at line 97, where we write "annual cycle" for the first time.

*10. Line 119: I am not sure the description on treatment of "hole" is complete that others can reproduce. Let 0 be non-blocking and 1 be blocking. What would the code say about 001010100, and 001110101011100?*

The code processes the "labels" resulting from k-means (i.e. 0, 1, 2, 3 for k=4) in the following order: 1) two blocking events longer than two days separated by a hole form one blocking event; 2) one blocking day and one blocking event longer than three days separated by a hole form one blocking event (the code checks first the case with order 11101 and then 10111). The code stops searching for the holes after checking the conditions 1 and 2.
The first example reported by the Referee (*001010100*) does not satisfy the previous conditions, therefore, all zeros remain holes. The second example (*001110101011100*) satisfies the second condition: *001110101011100 → 00111* **1** *101011100 → 0011111011* **1** *11100*. Therefore, the final result is: 2 blocking events of 5-day duration. We would like to reassure that these examples are extreme cases, as usually the k-means result (4530 labels, i.e. the number of days of 30 winters) does not present such an "unstable" sequence.
We modified the text to be more precise in the explanation at line 121:

"Therefore, the k-means result is processed in such a way that 1) two blocking events equal to/longer than two days separated by a hole form one blocking event and 2) one blocking event equal to/longer than three days and one blocking day separated by a hole (and then vice versa) form one blocking event."

11. *Line 125: Is "hole" included in "blocking days"?*

We call blocking days only those days which belong to a blocking event (which is defined in subsection 3.2). After the identification of blocking events, we do not speak about "holes" anymore but only about blocking days, although a blocking day could be an "old" hole.

12. *Line 136: I prefer a simpler phrase "non-zero" in place of "non vanishing".*
Done.

13. *Line 136: I would suggest to mention the 75m/100m threshold here in main text, rather than having to find it in supplement.*

We added the threshold values in subsection 3.3.

14. *Line 154: With Fig. 1, what is being evaluated is not ability to reproduce the "blocking weather regime" but "composites" (as defined in line 124-127).*
Done.

15. *Fig. 1: I assume this figure is based on DeltaZ500_HIST, so the overall higher geopotential height is included? From Fig. 3, I guess the overall Z500 increase is more than 25m in SSP2. Why don't I see an increase of RMSD because of this?*

There is an inaccuracy in the text, we thank the Referee to point it out. The Taylor diagram shows the central root-mean-square difference (CRMSD = $\sqrt{\frac{1}{N}\sum_i\left[(p_i - \bar{p}) - (r_i - \bar{r})\right]^2}$, where $p$ = prediction (GCMs) and $r$ = reference (ERA5)), therefore, the effect of the overall higher Z500 is not included in the results for SSP2 and SSP5. If we computed the usual RMSD for $\Delta$Z500$_{HIST}$ we would obtain, for example for MPI, 48 m in SSP2 and 82 m in SSP5, while the CRMSD values (in Fig.1) are: 18 m in SSP2 and 29 m in SSP5. The latter values are almost equal to the RMSD obtained for $\Delta$Z00$_{SSP}$. In the revised version of the manuscript, we showed only the results obtained with $\Delta$Z500$_{SSP}$ in the Taylor diagram and computed the usual RMSD.

16. *Table 1: The resolution of GFDL is said to be 1 degree on https://wcrp-cmip.github.io/CMIP6_CVs/docs/ Could the authors please check? I assume the argument made in line 179 is based on the resolution when the model is run, not the resolution of the output.*

We checked the source files stored in the Mesocentre ESPRI (in /bdd/CMIP6/CMIP/NOAA-GFDL/GFDL-CM4/historical/r1i1p1f1/day/zg/gr2/latest/) and we can confirm that the resolution of GFDL-CM4 is 2.5°x2.0° (as written in Table 1), with 12960 grid points for the globe (144 in longitude x 90 in latitude).

Yes, the sentence at line 179 refers to the model runs.

17. *Line 211/227: I am not sure the similarity between DeltaZ500_HIST and DeltaZ500_SSP is entirely interesting. The overall increase in geopotential height is a shift of all clusters in a hyper space. So by construction, it has no effect on the clustering result. The only difference is the seasonal variation around the mean. The similarity in results can only suggest the seasonal cycle does not alter enough from HIST to SSP to alter the clustering result.*

As mentioned before (at page 3), we showed the DeltaZ500_SSP results in the revised

version of manuscript and we moved most of the DeltaZ500_HIST results to the Supplement (or we removed them).

*18. Fig. 5: The peak of ERA5 at 27-28 days look suspicious.*

We checked the computations and the plot is correct. We obtain 1 blocking event of 27 days and 2 blocking events of 28 days (so we have three events in the 27-28 bin in Fig.5). This looks less suspicious if we look at the plots in Fig.S5: events that are longer than 20 days occur rarely in 30 years but it can happen to see a peak, like also for GFDL in HIST and for INM in SSP2.

*19. Line 247: Perhaps you can clarify "variability". Do you mean inter-model variability, or inter-event variability?*

We meant "inter-event" variability (however, this sentence is not present in the new version of the manuscript).

*20. Line 267: I don't think the 0.1% is a significant digit if the area only has two significant digits.*

Ok, we removed the percentage.

*21. Line 270: I think Nabizadeh et al. 2019 is based on DeltaZ500_SSP. And the increase you are talking is drastically larger than 17% in Nabizadeh et al. 2019. I don't know if I would call this agreement.*

This sentence is not present in the new version of the manuscript.

*22. Line 304-305: The sentence looks contradicting to me. ("may not match", "agrees well").*

We modified the sentence to "Despite the number of DG-blocking days may not match with the duration of the WTD-blocking events, we find that it generally agrees with the duration of the WTD-blocking events." This concept should be clear in the revised version thanks to the addition of the Figure included in the reply to Referee #2.

*23. Supplement Step A: I assume this step applies both to the center method and the DG method. But the step uses "blocking center". What is the "blocking center" for the DG method? Also, for the DG method, is there at most one such center/blob on each day, such that step 8 in the DG method only does temporal mean but not event mean?*

We thank the Referee to point this imprecision out, in fact, Step A applies to both methods but "blocking center" was defined only for the center method. In the revised version of the manuscript, we called "DG-grid boxes" those grid boxes where DG > 1.5 for at least five consecutive days and we corrected the explanation in Step A.

In the DG method there could be more than one blob per day, therefore, only the blob including at least one DG-grid box defines the blocking area of that DG-blocking day (as explained in Step A). The temporal mean of the areas of the DG-blocking days belonging to the same WTD-blocking event determines the area of the DG-blocking event.

---

## Author Comment (AC2)

**Authors' replies to Referee #2**

We really thank Referee #2 for their helpful suggestions and comments about our manuscript. Below, we provide our replies; the line numbers and sections refer to the old version of the manuscript.

**Major comments**

**Major suggestion #1**

*You need to provide a strong rationale for why you use the weather typing approach for identifying blocks as opposed to a feature-based method that allows simultaneous tracking in space and time. What are its advantages? I don't think that simply stating that others have used it before is sufficient. There are places in the manuscript where you say things that really make me wonder, why did they choose such an approach, such as: Line 261-262: "This is due to the fact that the events have been identified via a partitioning algorithm (k-means) and not via blocking indexes designed for geopotential fields that are typical during atmospheric blocking." You don't need to remove these sentences, but earlier, in the methods section, you need to make a convincing argument as to why this method useful. Given that there are multiple blocking indices that are designed for geopotential field, and that you felt compelled to compare your method against one of these, I want to know: why use the weather type approach?*

*Related to this, at line 340-341, you write: "Our results are in agreement with previous findings where blocking events are defined with blocking indexes. This confirms that the application of the WTD is also a good strategy to analyse blocking event characteristics." For me, this is a bit confusing. Why do we need another strategy that gives us the same information that we already have?*

We thank the Referee to point out these observations. We explained the reason for choosing the WTD in the Introduction (please, see reply 1a to Referee #1) and in subsection 3.1 at line 94: "Overall, while identifying blocking via blocking indexes implies making several choices, identifying blocking via the WTD can be considered as a standard procedure. This motivated us to apply the WTD and to explore this methodology for identifying blocking events and then studying their main characteristics (frequency, duration, size)."
Moreover, it is maybe worth reminding that the WTD application allows one to analyse more (in this case four) weather regimes at the same time, otherwise, one index per weather regime should be used. This could lead to some inconsistency, for example, the attribution of multiple weather types to the same day.

Regarding the second part of the comment of the Referee: since there is not one only way nor one best way to identify blocking (as explained in the Introduction) we carried out this study to explore if the WTD can be used to identify blocking events. Like the blocking indexes, this strategy has pros and cons and will not substitute the indexes, rather, it is another possibility which can be favored or not according to the purpose of the study.

Moreover, we added Figure 1 (below) in the Supplement to show the agreement between the results obtained with the center method and the DG method: 70%-85% of the blocking durations identified via the WTD overlaps (or coincides) with the durations identified via the DG method, showing a quite high correlation ($R > 0.74$). The underestimation of the DG-blocking duration is due to the fact they are identified within the WTD-blocking events (as

written at line 145).

[Figure]

Figure 1: Correlation between duration of WTD-blocking events and DG-blocking events during the HIST period. 70%-85% of the total blocking events identified via the WTD are shown; those events which are not identified with the DG method are not displayed.

**Major suggestion #2**

*Your choice to use shorthand names for the anomalies based on the climatology used is a bit confusing. For instance, when I first examined Figure 3, I thought there was a typo. Is there any chance that you might consider creating shorthand names that refer to both the removed and the dataset used, then, for instance: Z500_HIST composites during the winter HIST period (1980-2009) could be named: Z500_HIST_HIST. Whereas, Z500_HIST composites during the winter SSP2 period (2070-2099) could have the shorthand name: Z500_HIST_SSP2. This change would also help in clarifying what the x- and y-axes are referring to in Figure 4, are all of these showing results using the historical climatology, or is the x-axis the future climate with the historical climate removed? I eventually sorted out the answer to these questions, but with your existing format, it was more difficult than it needs to be.*

We thank the Referee for suggesting this new notation. We agree that referring to both the removed and the used datasets is more intuitive, so we followed this notation in the

new version of the manuscript. Actually, since we emphasized the results obtained with Z500_SSP2 and Z500_SSP5, we used the notation $\Delta Z500_{\text{SSP2}}$ and $\Delta Z500_{\text{SSP5}}$ for the anomalies in the future period (2070-2099) obtained by subtracting the climatology computed in the future; we thought that it is not necessary (and not nice) to write SSP2-SSP2 in subscript (i.e. $\Delta Z500_{\text{SSP2}-\text{SSP2}}$). Consistently, we kept the notation $\Delta Z500_{\text{HIST}}$ for the past. Instead, for the results obtained with Z500_HIST, we used the notation $\Delta Z500_{\text{SSP2}-\text{HIST}}$ and $\Delta Z500_{\text{SSP5}-\text{HIST}}$.

In the plots of Figure 4, the x-axis is for the results obtained in the HIST period (i.e. using $\Delta Z500_{\text{HIST}}$), while the y-axis is for the results obtained in the future using $\Delta Z500_{\text{SSP}}$ (i.e. $\Delta Z500_{\text{SSP2}-\text{HIST}}$ and $\Delta Z500_{\text{SSP5}-\text{HIST}}$). This type of plot wants to show the comparison between future and past in an easy way, as an alternative to reading the table. We improved the caption of Fig. 4 using the new notation.

**Minor comments**

*Line 99: possible typo, I think the word "so" should be "to".*
  Done.

*Line 130 and multiple places elsewhere: when you use the word extension, do you mean the same thing as extent? For me, extension suggests an action, such as expansion or shifting in the location of the block, whereas, extent suggests the instantaneous location of the block. I am curious to see if the other reviewers or the editor agree with me on this. If they do not, you can leave it as is.*
  We replaced the word "extension" with the term "size" in order to avoid any misunderstanding, as also suggested by Referee #1.

*Lines 131-135: This explanation did not make sense to me, mainly because you sometime mention composites and other times you do not, i.e., on line 132, you write: "The center method starts from the detection of the center of each blocking event." Okay, for me, because you say "for each blocking event", I think to myself: this does not involve composites. But then the next sentence talks about defining the centers based on the anomaly for the composite. Please try to re-write this description to remove any confusion.*
  We reviewed the usage of the expressions "composites" and "blocking events" and we modified the text according to the Referee's comment.

*Line 186: I think Figure S3 should be included in manuscript as a result figure not a supplemental figure. In the current form of the manuscript, a comparison of Figures 2 and 3 gives a strong suggestion that blocking changes significantly with climate change. But that is not the result that you are presenting. Instead, the differences in Figures 2 and 3 is mainly a due to a difference in the mean state for the 20 and 21 st centuries. Right? If you see it differently, then please explain.a*
  Following also the suggestions of Referee #1, we showed the DeltaZ500_SSP results in the revised version of the manuscript and we moved most of the DeltaZ500_HIST results to the Supplement (or we removed them). Thus, also Figures 2, 3, and S3 changed as asked by the

Referee.

*Lines 252-257: This discussion of block intensity when considering delta-Z500HIST vs delta-Z500SSP is a bit awkward for me. Given that in the present climate we define blocks as anomalies with respect to the climatology, your discussion of future block "intensity"relative to the historical climatology, seems a bit arbitrary. For context, if we were discussing heat waves, we know that humans feel uncomfortable when the temperature is above a certain threshold, e.g. $32°C$. So, when someone studies future warm events, there is a reason to look at anomalies with respect to our current climatology. But for block intensity, does that same thing hold? i.e., when surface pressure is above some threshold are there specific impacts on humans? If so, please explain. If not, then I wonder if you might consider streamlining this section and only discussing the intensity changes when comparing delta-Z500HIST for the historical runs with delta-Z500SSP for the future runs. Especially since your discussion of Figure 3 already makes the point about the impact of the change in the climatology on the Z500 anomalies.*

Please, see our previous reply.

*Figure 7, one result you have found, that I don't think is mentioned (pardon me if I missed it), is that the model-to-model differences in intensity are larger that the intensity differences for SSP2 vs SSP5. This suggests to me that even though the models are getting some aspects of the physics of blocking correct, there is room for improvement.*

The fact that the model-to-model differences are larger than SSP2 vs SSP5 was mentioned at line 209, but we added this observation also at line 245. This could suggest that the signal of climate change on this particular quantity (blocking frequency/duration) is not significant and that taking a stronger warming scenario does not change much the results; we commented this in the Conclusions.

*Line 331, you write: "Climate change will significantly increase the extension of blocking events in the future especially in the worst-case scenario." I don't think your results agree with this statement. As you discuss, the difference is related to a change in the climatology. For me, this implies that the blocks themselves, in terms of their size and impact on the circulation, will be similar to what we observe in the current climate. Perhaps a method for testing this would be to address the question: are the pressure gradients associated with the blocks strengthening?*

As we showed the DeltaZ500_SSP results in the revised version of the manuscript, as written before, this sentence is not present in the Conclusions.

*Philosophical Suggestion: Whenever a "trends" manuscript is written, I think the authors should take time to ponder the question: Are we (or somebody else perhaps) going to have to write this same paper in N years when the next generation of climate models are released? If the answer is yes, then why do we need this paper now? If the answer is no, or not exactly, then why not – what have we established here that is robust to potential changes in models? And maybe some elements will be updated, but are there are least some elements that are novel in this study that are not going to be superseded by results from an improved set of climate models, and/or are the hypotheses or theories developed in this worak that provide a simpler guide to how we can interpret the trends?Obviously, I have not reviewed the majority of trends papers written, so most of the time this question is not addressed. So, I won't hold you to a higher standard than what exists in the literature. However, I encourage you to consider*

*engaging in this activity and perhaps including a paragraph at the end of the paper addressing this issue. No problem if not.*

Investigating future blocking with the last generation of models is motivated by the fact that controversial results have been found so far about future blocking and CMIP6 models were shown to better represent the blocking weather type. It would be natural to propose again such an investigation if the previous conditions persist. At the same time, we would like to stress that in this study we introduce a different methodology to study spatio-temporal characteristics, showing that this can be an alternative to the blocking indexes; this methodology can be reused in future studies with newer generations of models.

---

## Author Response (AR2)

**Dear Editor,**

thank you for your comments and suggestions. In this document, first we reply to your two comments, then we report the modifications that we implemented in the new version of the manuscript.

L53-55: it would be helpful to add a few more words and mention that Hassanzadeh et al's paper showed a decrease in blocking activity with Arctic warming or just that it highlighted the need for more study on this link, or something along these lines.

We modified the sentence; please, see our reply to Referee #2.

Please carefully address the reviewer's comment 1 and their other comments/questions. You might also want to take a look at this paper by Falkena et al. https://rmets.onlinelibrary.wiley.com/doi/full/10.1002/qj.3818. Based on these comments and these studies, I suggest that you consider taking a look at the potential effects of increasing k (e.g., to 6).

We thank the Editor for this interesting reference, which we cited in the manuscript. Falkena et al. (2020) found that k=4 is the best choice when k-means is applied on the reduced data, while they showed that k=6 is the optimal choice when k-means is applied on the full field data. Since it is well documented that k=4 is a good choice when k-means is applied after PCA, we did not repeat the analysis for k=6. Please, see our discussion about k in the reply to comment-1 of Referee #1.

The major changes in the last submitted manuscript are in the Conclusion section (now named "Conclusions and discussion"), where we added some discussions and comments about the points raised by the referees. Small changes are present in the Introduction and in the presentation of the results (sections 4.3 and 4.4), as we removed two plots regarding the  $\Delta Z500_{SSP2-HIST}$  and  $\Delta Z500_{SSP5-HIST}$  results and reorganized the text and the figures.

These changes and all the smaller corrections are visible in the marked-up version of the manuscript.

Thank you very much!

Yours faithfully,

Sara Bacer

**Authors' replies to Referee #1**

We really thank the referee for the interesting comments; they were helpful to improve the manuscript. Below, we provide our replies.

**Major comments**

1. WTD. After my first review, I came across Dorrington and Strommen 2020 (hereafter DS2020) and Dorrington et al. 2021, which answered some of my questions about WTD.

Fitness of WTD: DS2020 found that the raw Z500 phase space (before removing the jet speed variability) is quite Gaussian (DS2020's Fig 1a). In other words, the clustering is not very clear cut. DS2020 found the standard/classical k=4 clustering to be problematic. "When setting K=4 in our residual space, we found k-means clustering consistently returns Clusters 3, 4, and 5 of Figure 4 but in different 30-year windows switches between including Clusters 1 and 2" (quoting from last paragraph of section 3.4 in DS2020). Such decadal variability in cluster centroid was considered not a real signal, but artifact of the choice of k=4.

Interpretation of WTD: Back to your work, your response to my previous comment 1c did not address my questions. (Maybe I was not clear enough.) I still cannot understand what is different in the \*raw model output\*, that causes difference in the WTD analysis. Part of my questions relate to your choice to allow cluster centroids to differ and that the clustering is not clear cut. DS2020 made the clustering clear cut by removing the jet speed variability, and they worked towards requiring "dynamically relevant regimes to be approximately stationary features of the midlatitude circulation over centennial time scales, at least in terms of spatial patterns if not in residence times or transition probabilities" (quoting from section 3.2 in DS2020).

When you find "cluster centroid" to be different in some models, could it be the models behave like different 30-year windows in the example of DS2020? And that difference gets overly exaggerated because of the bad choice of k=4?

On the interpretation of "blocking frequency" and "blocking center", I would like to see the output of the WTD clustering (the weather type and its frequency) as an intermediate step before the "blocking frequency" and "blocking center". I think that helps readers interpret the results. Meanwhile, fundamentally, I am looking for more clear-cut clustering, or to use the same cluster centroid. In this way, I can better trust that WTD clustering can faithfully summarize the Z500 variability, and will not overly exaggerate noises.

**We thank the referee for these interesting recent references.**

We would like to point out the following considerations.

- In DS2020, the analysis is repeated for different 30-year windows along the 20th century, e.g. Fig.2b. Since their aim is to perform a statistical study of criteria such as the BIC, the analysis does not show the results for each 30-year period explicitly. Therefore, we cannot know if the results for the period 1902-1931, for instance, are close or not to the results for the period 1912-1934 (i.e. shifted by 10 years).
- In our study, instead, we consider only two 30-year periods in order to study the difference between the end of the 21th century and the past.

- There is no rule about how many years to consider for the WTD; for instance, Cassou 2008 considered 33 years and Ullmann et al. 2014 considered 26 years.
- If we considered 40 years or other 30 years shifted by 10 years (e.g. 1970-1999 for the past and 2060-2089 for the future) and if we obtained results slightly different from the ones in the manuscript (but still not significant changes between past and future), this would mean that blocking changes are small and within the climate intra- and inter-decadal variability, like suggested in Huguenin et al. (2020). We added some comments at L320 in the conclusions (please, see the end of this reply).
- The differences that we find in terms of frequency and persistence of weather types across the models are not new and have already been discussed (e.g. Huguenin at al. 2020).

Therefore, we do not expect that the results of our analysis depend on the selected 30-years.

Considering the published literature, we do not think that "k=4 is a bad choice". k=4 has been widely documented to be the best choice for the classification of winter weather types in the Atlantic sector, and such value of k has been largely used in other studies (please, see the citations in section 3.1 in the manuscript and references therein). It is true that, recently, some studies suggested other values of k, but these findings must be contextualized (e.g. on which data set exactly is k-means applied?) and confirmed by further studies. We write below two examples.

- Falkena et al. 2020 found that k=6 is the optimal choice when clustering is applied on the full field data. However, when the clustering is applied on the reduced data, i.e. the PC data, they found that the optimal number of clusters is 4.
- DS2020 found that k=2, 5, or 6 is a better choice using raw PCs, while k=3, 5, or 6 is better when using residual PCs, i.e. after removing from the geopotential height the influence of the jet speed. The first result is clearly in contrast with a large number of previous studies (all those for which the best choice is k=4), but we could not find any explanation for this disagreement in the paper.

Moreover, we observed that DS2020 retained only 10 PCs to explain about 83% of the variance. However, the number of PCs can influence the choice of the best value for K. Indeed, Falkena et al. (2020) showed that "For lower numbers of EOFs, the optimal number is found to be lower, while a higher number of EOFs leads to a higher optimum for k. This is to be expected because the use of a limited number of EOFs means that some variability of the original data is neglected. This loss of variability is larger when fewer EOFs are used, and as a consequence fewer clusters are needed to account for the variability of the EOF data." In particular, Falkena et al. found that k=4 is the best choice using 20 PCs, K<4 using 10 PCs.

Overall, we believe that, for our paper, k=4 is not a bad choice and is a recommended one as it is well justified in the literature.

We added at the end of this document the four centroids (for the four weather types: blocking, Atlantic ridge, NAO+, and NAO-) and the frequency of occurrence obtained for all models (for the historical period).

As commented in our replies in the previous report, k-means does not perform a clear-cut clustering. However, our methodology relies on a well established and largely used approach to define weather types: PCA+clustering. Moreover, we find similar results for the size of the blocking area by applying the WTD method and the DG method (which uses the DG index on the raw data), see Figure S6. This suggests that the noise associated with the non clear-cut clustering does not dominate the results.

Finally, we would like to add that, before adopting this procedure, we followed the approach of using reference eigenvectors and reference centroids (i.e. the ones obtained with the reanalysis) to find the weather types of the GCMs (like in Ullmann et al. 2014). However, the blocking pattern obtained with some GCMs did not resemble the reference blocking pattern; instead, by using the PCA + k-means approach for each GCM, we have always obtained blocking regimes close to the reference blocking pattern.

L320 (Conclusions): "Given the decadal variability of weather regimes (Dorrington et al. (2020), longer past and future periods could be considered (e.g. periods of 50 years, like in Fabiano et al. 2020) so as to better smooth the dependency of the results on this decadal variability. Moreover, those days for which the geopotential height anomaly field does not resemble the blocking weather regime pattern could be classified as "neutral days", like in Dorrington et al. (2021), and excluded from the analysis."

L324 (Conclusions): "The optimal number of clusters depends also on the data to be processed; for instance, by applying the clustering on the full field data, Falkena et al. (2020) found that k=6 is an optimal choice."

**Minor comments**

2. Insignificant results. Related to my previous comment 1d. Now, after taking away results using DeltaZ500\_HIST, most results are not statistically significant. If you cannot exclude that non-clear-cut clustering might have a role in making the results insignificant, I think you should acknowledge that in your manuscript.

While I agree that "Finding not statistically significant changes is a result itself", I encourage authors to cite papers that contrast with the results here. Huguenin et al. 2020 also used some kind of circulation type classification, and found lack of change in frequency and persistence, but they cited papers which contrast with their results. You may also read Kautz et al. 2021 and Nabizadeh et al. 2021, where you may find some discussion on blocking under climate change.

We added at L320 (Conclusions) the following sentence: "Finally, a sensitivity analysis of the results to the clear-cut character of the clusters could be conducted, for instance by removing the influence of the jet speed from the geopotential height field, like in Dorrington and Strommen (2020)."

We thank the referee for these interesting references. We changed/added some text in the Introduction and Conclusion (this section is called "Conclusions and discussion" now) to improve our discussion on the impact of climate change on blocking frequency, duration, and size and compare better our results with previous findings (which can agree or not with ours). (Blue parts are the main changes.) L55-64 (Introduction): "So far, studies have mainly focused on frequency and duration of future blocking events. Some of these studies found that blocking frequency will decrease in the Northern Hemisphere (e.g. Dunn-Sigouin and Son, 2013; Matsueda and Endo, 2017; Fabiano et al., 2020, Davini et al. 2020), while blocking duration may either increase (Sillmann et a. 2009) or decrease (Fabiano et al. 2020). Other studies showed that blocking frequency and duration will not change notably in warming climate (Dunn-Sigouin and Son, 2013; Huguenin et al. 2020). Future changes in blocking size have received less attention (Hassanzadeh et al. 2014; Nabizadeh et al., 2019).

Most of the studies mentioned above determined blocking events via blocking indexes (Sillmann and Croci-Maspoli, 2009; Dunn-Sigouin and Son, 2013; Hassanzadeh et al., 2014; Matsueda and Endo, 2017; Nabizadeh et al., 2019; Davini and D'Andrea, 2020) and considered GCMs participating in the Coupled Model Intercomparison Project phase 5 (CMIP5 Dunn-Sigouin and Son, 2013; Matsueda and Endo, 2017; Huguenin et al., 2020) or idealized GCMs (Hassanzadeh et al., 2014). To our knowledge, only Fabiano et al. (2020) employed CMIP6 models in order to project the blocking weather type and analyse its changes in frequency and duration in the 21st century."

L304-317 (Conclusions and discussion): Please, see directly the new version of the manuscript.

3. Line 14-15: You may also refer to Kautz et al. 2021 [doi:10.5194/wcd-2021-56]. We added this citation.

4. Line 124: Consider change "net impact" to "total impact" or "gross impact". We used the expression "gross impact".

5. Section 3.3: Your response to my previous comment 10 helps a bit, but the revised manuscript is still not clear on the treatment of holes.

*Line 128. "longer than five days"->"at least five days long"*

Line 129. Remove "and separated by at least two non-blocking days". Because this is not true for the 2nd example I gave last time (001110101011100).

Line 129. Consider change "is assumed to represent"->"might represent", in order to soften this sentence, because this is not true for the two examples I gave last time.

Line 130. Consider change "Therefore" to "Concretely", because this sentence is what the code does, not only examples.

Below are more subtle details. One way is that you can reply here and refer to the discussion here in the manuscript. If you are doing find-and-replace in place, the searching order of (11011,11101,10111) matters, e.g., 00110110100, 00111010100. Would be good to make explicit the order of searching. Overlapping matches can be bad for codes, e.g., 110111011 is two 11011 overlap together. Does your code find one or two matches of 11011? What will your code say about 001101101110100?

We changed the sentences at lines 128, 129, and 130, as suggested by the referee.

First of all, we would like to point out an inaccuracy in our reply to the comment-10 of the previous report: the second example provided by the referee in that comment (i.e. 001110101011100) gives 00111111111100 (i.e. one blocking event that is 11 days long).

A different result is obtained with a similar example:  $0010101011100 \rightarrow 001010111100$  (i.e. one blocking event that is 5 days long). In this case, the vice versa of the condition (2) (i.e. "one blocking day and one blocking event equal to/longer than three days separated by a hole") is verified once.

We write below our algorithm's results for the examples asked by the referee.

 $00110110100 \rightarrow 00111111100$ : first condition (1) and then condition (2) are applied;  $00111010100 \rightarrow 001111111100$ : condition (2) is applied twice;

 $0011011101110100 \rightarrow 001111111111100$ : first condition (1) is applied twice, then condition (2) is applied.

We would like to reiterate that the modifications of the sequence (i.e. the suppression of holes via the conditions (1) and (2)) are very rare: the sequence of labels (4530 elements) is modified 13 times on average, i.e. around 0.3% of the length of the sequence. Moreover, we would like to point out that the condition (2) is verified less than 30% of the times. We added one sentence in the manuscript to stress this aspect: "Overall, the number of holes that are converted into blocking days is very small, about 0.3% of the number of winter days (4530)." In this way, we would like to reassure that the "questionable" cases are few and cannot affect our results.

6. Fig. 1 caption: Related to my previous comment 15, how about you say in the caption that it is the CRMSD, not RMSD?

Writing RMSD in the caption is correct. In fact, our current Fig.1 shows the RMSD values computed with  $\Delta Z500_{HIST}$ ,  $\Delta Z500_{SSP2}$ , and  $\Delta Z500_{SSP5}$ , while the old Fig.1 (i.e. the one in the manuscript of the first submission) showed the CRMSD values computed with  $\Delta Z500_{HIST}$ ,  $\Delta Z500_{SSP2-HIST}$ , and  $\Delta Z500_{SSP5-HIST}$ . We are sorry if we were not clear in our previous reply to comment-15.

7. Supplement Step A: Related to my previous comment 23, can there be more than one blobs that contain a DG-grid box? How are they treated?

It rarely happens that DG-grid boxes form different "blobs" and identify different blobs of  $\Delta Z500$ , like Day 1 in Figure 1. If this happens, since the "scan" of the domain by the algorithm starts from the top-left (or North-West) corner, only the first encountered blob (between 30W and 50E) is retained. On Day 1, only the blob over Scandinavian is retained (last row in Figure 1), however, since the next day, any "blob of DG-grid boxes" (second row) detects the same blob of  $\Delta Z500$ . We observed that this is the typical situation in those rare cases in which there is more than one "blob of DG-grid boxes".

Overall, most of the times there is only one "blob of DG-grid boxes" (or even only one DG-grid box) per day and, when there are two "blobs of DG-grid boxes", these usually identify the same blob of  $\Delta Z500$  (like Day 2 and Day 3 in Figure 1); the case of Day 1 is possible but unlikely.

---

## Author Response (AR3)

Dear Editor,

please, find our replies to Referee#1 at pages 2 and 3 of this document.

We implemented the two minor corrections required by the referee at pag. 2 and pag. 14 of the manuscript (visible in the latexdiff document).

We also added a new affiliation for the second author.

Finally, we would like to inform you that we changed the colors in the figure of the manuscript using color schemes accessible to persons with color vision deficiencies, as required by the journal.

Thank you very much!

Yours faithfully,

Sara Bacer

**Authors' replies to Referee #1**

We thank the referee for the interesting and helpful comments. Below, we provide our replies.

**Minor comments**

*1. **Value of k in WTD.** I like your critical reading of DS2020---to the extent that I can accept that there are multiple alternative values of k proposed recently and consensus are yet to be reached. I do have reservations to your other interpretations of DS2020 (e.g., their Fig.2b). But because I am not further questioning your choice of k, further discussion of DS2020 might not help much in assessing your manuscript. And thanks to editor for bringing up Falkena et al. (2020), which definitely help the discussion of the value of k, etc.*

We thank the referee for the comment and for having stimulated this interesting discussion.

*2. **Results using reference centroids.** In your response, you said if reference centroids (obtained from reanalysis) are applied to the GCMs, "the blocking pattern obtained with some GCMs did not resemble the reference blocking pattern". I am surprised by this. Are you sure?*

We did not apply this method (i.e. using the reference centroids to define the GCM centroids) to all GCMs used in our study, but we applied it to two GCMs. Figure 1 (below) displays the difference between the centroids obtained with this method and the ones obtained by applying the WTD on the GCM Z500 anomaly field (i.e. the methodology applied in our manuscript). In this example, the MRI-ESM2 model is considered: only Atlantic Ridge is well captured using the reference principal components (PCs) and the reference centroids.

[Figure]

Figure 1: Centroids obtained with the reanalysis (left) and with the MRI-ESM2 model in two different ways (center and right).

*3. Some GCMs less accurate in capturing blocking? In your response, you said "using the PCA + k-means approach for each GCM, we have always obtained blocking regimes close to the reference blocking pattern." But in your manuscript, line 182, you said "MIROC, CanESM, and IPSL are less accurate in capturing the blocking pattern". This is contradicting.*

The two sentences look contradicting but they refer to two different methodologies. We said that *"by using the PCA + k-means approach for each GCM, we have always obtained blocking regimes close to the reference blocking pattern"* in comparison to the method commented in the previous reply (i.e. using the reference centroids to define the GCM centroids). On the

other hand, we wrote in the manuscript that "*MIROC, CanESM, and IPSL are less accurate in capturing the blocking pattern*" with respect to the other GCMs when the WTD is applied to all GCMs. Graphically, we mean that the blocking composites for MIROC, CanESM, and IPSL in Fig.2 in the manuscript are less accurate than the other GCMs in Fig.2, but they are anyway closer to the typical blocking pattern than the blocking centroid obtained in Figure 1 (right) of the present reply.

Overall, what is written in the manuscript is consistent, and the sentences cited by the referee look contradicting because they refer to two different situations.

*4. "Some GCM less accurate in capturing blocking" is perhaps the only significant result in the paper. But I raised questions on this in my previous review. Not sure if you responded, so I rephrase it here again:*

*In assessing the robustness (or "stationarity") of clustering methods: DS2020 found largely different cluster centroid in different 30-year windows; Falkena et al. (2020) said "The differences between the results for odd and even years are found to be large". Both were considered not a real signal, but a behaviour of lack of robustness.*

*When you find "cluster centroid" to be different in some models, could this be an outcome of lack of robustness of the clustering methods?*

We believe that our method is robust as claimed in our previous report, in the reply to comment-1, where we wrote that "*our methodology relies on a well established and largely used approach to define weather types: PCA+clustering. Moreover, we find similar results for the size of the blocking area by applying the WTD method and the DG method (which uses the DG index on the raw data), see Figure S6. This suggests that the noise associated with the non clear-cut clustering does not dominate the results.*" Hence, the differences among models is not due to a lack of robustness of the methodology but to the inter-model uncertainty (see also Huguening et al. 2020).

The sentence "the differences between the results for odd and even years are found to be large" of Falkena et al. 2020 refers to the test done using half of the data set (i.e. 39/2 years), and they suggest that such a data set could be not "of sufficient length to draw reliable conclusions about the clustering results".

Finally, regarding the DS2020's work, as we already commented in our previous report, in the reply to comment-1, we do not know how large the difference between the results obtained for two 30-year windows shifted by only 10 years is. Indeed, DS2020 shows the differences among the clusters along the 21st century, but this is not directly comparable to our study.

*5. Line 57: "further investigations are necessary to define the Arctic amplification response to blocking". Do you mean "further investigations are necessary to understand the Arctic amplification effect on blocking"?*

Yes, we changed the sentence.

*6. Line 302: Consider change "last generation" to "latest generation".*

Yes, thanks.